# Maximal power per device area of a ducted turbine

Nojan Bagheri-Sadeghi[1], Brian T. Helenbrook[1], and Kenneth D. Visser[1]

[1]Department of Mechanical & Aeronautical Engineering, Clarkson University, Potsdam, NY 13699-5725, United States

**Correspondence:** baghern@clarkson.edu

**Abstract.** The aerodynamic design of a ducted wind turbine for maximum total power coefficient was studied numerically using the axisymmetric Reynolds-averaged Navier-Stokes equations and an actuator disc model. The total power coefficient characterizes the rotor power per total device area, rather than the rotor area. This is a useful metric to compare the performance of a ducted wind turbine with an open rotor and can be an important design objective in certain applications. The design variables included the duct length, the rotor thrust coefficient, the angle of attack of the duct cross-section, the rotor gap, and the axial location of the rotor. The results indicated that there exists an upper limit for the total power coefficient of ducted wind turbines. Using an Eppler E423 airfoil as the duct cross-section, an optimal total power coefficient of 0.70 was achieved at a duct length of about 15% of the rotor diameter. The optimal thrust coefficient was approximately 0.9, independent of the duct length and in agreement with the axial momentum analysis. Similarly independent of duct length, the optimal normal rotor gap was found to be approximately the duct boundary layer thickness at the rotor. The optimal axial position of the rotor was near the rear of the duct, but moved upstream with increasing duct length, while the optimal angle of attack of the duct cross-section decreased.

## 1 Introduction

The power output of a wind turbine can be augmented by surrounding it with a duct, typically referred to as a ducted wind turbine (DWT), a diffuser augmented wind turbine or a shrouded wind turbine. The effect of the duct is to increase the mass flow rate through the rotor. For a given rotor area, significantly more power can be obtained for a DWT compared to an open wind turbine. However, by adding a duct, the total area of the device facing the wind direction is increased. If the power produced per total projected frontal area of the device is calculated for DWTs, often values closer to that of open wind turbines are found (van Bussel, 2007). When nondimensionalized by the kinetic power available in a unit area of freestream, the power per rotor area and per total area of the DWT are referred to as rotor and total power coefficients and are designated by $C_P$ and $C_{P,total}$ respectively. For an open rotor turbine, these two power coefficients are equal. Achieving values of $C_{P,total}$ greater than the Betz-Joukowsky limit (Okulov and van Kuik, 2012) of 0.593 for a DWT is significant as it means a DWT can capture more power per unit area of the device than an open rotor turbine. Optimization studies of DWTs have shown that DWTs can achieve values of $C_{P,total}$ beyond the Betz-Joukowsky limit (Aranake and Duraisamy, 2017; Bagheri-Sadeghi et al., 2018). $C_{P,total}$ not only is a useful metric to compare DWTs with open rotor wind turbines but could be an important design objective

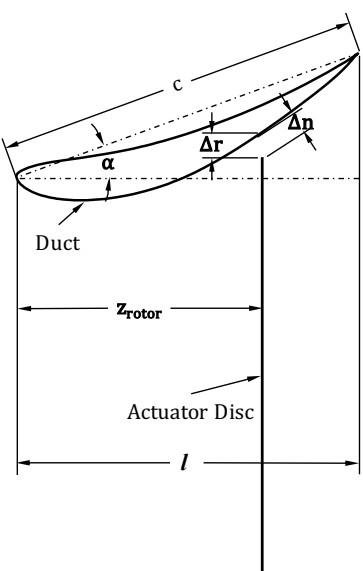

**Figure 1.** The design variables of a ducted wind turbine

in certain problems like fully-integrated DWTs for sustainable buildings (Ishugah et al., 2014; Agha and Chaudhry, 2017) or other applications where a designer seeks to maximize the power output from the limited space allocated to a wind turbine.

Since the experimental demonstration of the power augmentation provided by shrouding wind turbines (Sanuki, 1950), numerous studies on the design and optimization of the DWTs have been carried out (Lilley and Rainbird, 1956; Igra, 1981; Loeffler, 1981; Gilbert and Foreman, 1983; Georgalas et al., 1991; Politis and Koras, 1995; Hansen et al., 2000; Phillips et al., 2002; Ohya and Karasudani, 2010; Aranake and Duraisamy, 2017; Venters et al., 2018; Bagheri-Sadeghi et al., 2018). Only a few (Aranake and Duraisamy, 2017; Bagheri-Sadeghi et al., 2018; Venters et al., 2018) have used $C_{P,total}$ as a design metric while most other studies have focused on maximizing $C_P$. The design variables of a DWT include the rotor blade design, its axial location, $z_{rotor}$, and tip clearance or rotor gap, $\Delta n$, and the angle of attack, $\alpha$, length, $l$, and shape of the duct cross-section. These design variables can be seen in the schematic shown in Fig. 1 with the rotor replaced by an actuator disc.

Many of the numerical studies use axisymmetric CFD models (Loeffler, 1981; Georgalas et al., 1991; Politis and Koras, 1995; Phillips et al., 2002; Aranake and Duraisamy, 2017; Venters et al., 2018; Bagheri-Sadeghi et al., 2018). If the rotor blades are modeled as an actuator disc, the thrust coefficient can be considered a design variable, and different rotor loadings can be represented by different values of the thrust coefficient (Loeffler, 1981; Hansen et al., 2000; Phillips et al., 2002; Venters et al., 2018; Bagheri-Sadeghi et al., 2018). Similarly, some experimental studies replace turbines with screens of different porosity to study DWTs at various rotor loadings (Igra, 1981; Gilbert and Foreman, 1983). Most simplified theoretical models of DWTs indicate that the optimal power output is achieved at a thrust coefficient of 8/9 similar to open wind turbines (van Bussel, 2007;

Jamieson, 2008; Bontempo and Manna, 2020). Bontempo and Manna (2020) reviewed various theoretical models of ducted wind turbines and concluded that they are all equivalent and that their apparent differences are due to different choices of flow

parameters used to characterize the effect of the duct (e.g. the exit pressure coefficient (Foreman et al., 1978) or extra back pressure velocity ratio (van Bussel, 2007)). They also show that these models are insufficient in predicting the optimal design of ducted turbines as they neglect the dependence of the flow parameters on the thrust coefficient. However, they demonstrated that $C_{P,total}$ greater than Betz-Joukowsky limit can be achieved at $C_T = 8/9$. Bagheri-Sadeghi et al. (2018) reported an optimal value of thrust coefficient close to this value when optimizing for $C_{P,total}$ using Reynolds-averaged Navier-Stokes (RANS)

simulations with an actuator disc model. However, McLaren-Gow (2020) performed axisymmetric inviscid simulations of DWTs with various duct shapes with an actuator disc and concluded that the value of $C_T$ to maximize $C_P$ is lower than 8/9.

    As most studies focus on maximizing $C_P$, with a few exceptions (Georgalas et al., 1991; Politis and Koras, 1995; Venters et al., 2018; Bagheri-Sadeghi et al., 2018), the axial location of the rotor is usually fixed at the smallest cross-section of the duct where the maximum velocity is assumed. In our previous paper (Bagheri-Sadeghi et al., 2018), we included the axial position

of the rotor as a design variable and compared the optimal designs for maximum $C_P$ and $C_{P,total}$. We observed that the optimal axial location of a rotor to maximize $C_P$ or power is close to the duct throat. However, when optimizing for maximum $C_{P,total}$, the optimal axial position moves further downstream of the duct throat, which for a given rotor size results in a significantly smaller total area of the device.

    The effect of the angle of attack of the duct cross-section has been included in most studies (Georgalas et al., 1991; Politis

and Koras, 1995; Aranake and Duraisamy, 2017; Venters et al., 2018; Bagheri-Sadeghi et al., 2018). The results of these studies show that power output is considerably sensitive to the angle of attack of the duct cross-section and that more power is obtained by increasing the angle of attack of the duct cross-section up to the point where the flow separates inside the duct. As noted by Bagheri-Sadeghi et al. (2018), the optimal design of a DWT is on the verge of flow separation which is often accompanied by a sharp decrease in the power output. Therefore, the accuracy of CFD simulations significantly depends on the

accurate prediction of flow separation. The $k - \omega$ SST turbulence model (Menter, 1994) is more accurate than the $k - \epsilon$ models in prediction of the flow separation for flows with adverse pressure gradients and thus is preferred when RANS simulations are used to study DWTs (Hansen et al., 2000; Bagheri-Sadeghi et al., 2018). This sharp drop in the power which results in a discontinuity in the objective function has implications in the choice of optimization method too, as it renders methods assuming a smooth objective function ineffective (Bagheri-Sadeghi et al., 2018).

There are a few studies on the optimization of the shape of the duct cross-section for optimal $C_{P,total}$ such as Aranake and Duraisamy (2017) but the design space is limited by fixing some other design variables. For instance, Aranake and Duraisamy (2017) use a penalty function to avoid large values of the thrust coefficient, and the axial location of the rotor, the chord length of the duct cross-section, and the rotor gap were not introduced as design variables. In most studies, a high-lift airfoil with the suction side inside the duct is used as the cross-section of the duct (de Vries, 1979). A high-lift airfoil shape creates circulation

and thereby increases the mass flow rate through the duct. Further increases in $C_{P,total}$ could be possible by delaying boundary layer separation, e.g. by using multi-element or slotted ducts (Igra, 1981; Gilbert and Foreman, 1983; Phillips et al., 2002; Hjort and Larsen, 2014), as the optimal design tends to be on the verge of flow separation (Bagheri-Sadeghi et al., 2018). Large

flanges are often used at the exit of ducted turbines to further lower the pressure at the exit plane of the duct and increase the swallowing effect. Limacher et al. (2020) showed that large flanges lead to reduced values of $C_{P,total}$.

The effect of the rotor gap as a design variable is considered in a few studies (Politis and Koras, 1995; Venters et al., 2018; Bagheri-Sadeghi et al., 2018), and although optimum rotor gaps have been examined (Bagheri-Sadeghi et al., 2018), no conclusions about the optimal rotor gap for optimal $C_{P,total}$ have been obtained to our knowledge. Lastly, the effect of the duct length for a given rotor diameter on the optimal design of a DWT is only considered in a few works (Georgalas et al., 1991; Politis and Koras, 1995; Ohya and Karasudani, 2010; Venters et al., 2018). Within the range of duct lengths that seem practical (lengths smaller than the rotor diameter), studies suggest that $C_P$ can be increased monotonically by increasing the duct length. However, we are not aware of any studies on the effect of the duct length on the optimal design for $C_{P,total}$.

This study investigates the effect of the duct length on the optimal design for maximizing the total power coefficient, $C_{P,total}$, of a DWT having the Eppler E423 airfoil as the cross-section. This entails identifying whether there is an optimal duct length and how the optimal design variables of a DWT change as the duct length varies. The results show, that there is an upper limit to $C_{P,total}$ for a DWT, which is similar to the Betz-Joukowsky limit for open rotors. The result established here is specific to the Eppler E423 used for the duct cross-section, but a similar result should hold for other duct cross-sections as well. Additionally, the results indicate that the optimal rotor gap is close to the boundary layer thickness at the rotor independent of the duct length. Furthermore, this paper illustrates how the optimal axial position and the angle of attack of the duct change with increasing duct length.

The paper is organized as follows: section 2 discusses the CFD model used for RANS simulations including a validation study of the actuator disc model used followed by the details of the DWT design parameterization and the optimization method used. The optimization results and how the optimal design changes with the duct length are discussed in the third section. This section also involves a sensitivity analysis of $C_{P,total}$ of the optimal design, and a comparison of the power per unit device area vs. rotor thrust of the optimal DWT and an open rotor.

## 2 Methods

Ansys Fluent 17.1 was used to solve the incompressible RANS equations with the $k-\omega$ SST turbulence model (Menter, 1994). The computational domain used is shown in Fig. 2 which extends to $\max(4D, 15c)$ upstream of the rotor, and $\max(8D, 25c)$ downstream of it, where $\max(x,y)$ is the greater of $x$ and $y$, $D$ is the rotor diameter and $c$ is the chord length of the duct cross-section. The flow was considered axisymmetric and the rotor was modeled as an actuator disc. The pressure drop across each cell of the actuator disc was:

$$\Delta p = \frac{1}{2}\rho V_z^2 C_{T,rotor} \tag{1}$$

where $\rho$ is the air density and $C_{T,rotor}$ is the thrust coefficient based on the axial velocity, $V_z$, at the rotor. The value of $C_{T,rotor}$ was defined in the fan boundary condition of Ansys Fluent. Using $C_{T,rotor}$ as a design variable means that the pressure drop across the actuator disc was not constant. However, as it is easier to interpret, all the results are given in terms of

the thrust coefficient defined as $C_T = \frac{T}{\frac{1}{2}\rho V_\infty^2 A_{rotor}}$, where $T$ is the thrust force on the rotor, $V_\infty$ is the freestream velocity and $A_{rotor}$ is the swept area of the rotor. At the inlet, the freestream values of the turbulence variables were set as $\omega_\infty = \frac{5V_\infty}{D}$ and $k_\infty = \nu\omega_\infty \times 10^{-3}$ where $\nu$ is the kinematic viscosity of air. The flow field of RANS actuator disc simulations are sensitive to freestream values of turbulence variables (Bagheri-Sadeghi et al., 2020) and the values selected correspond to recommended values by Menter (1994). Ansys Fluent uses a cell-centered finite volume method. The pressure-based solver with the coupled algorithm and Fluent's second-order accurate schemes were used for all flow variables. The values of rotor power output and thrust were monitored to ensure iterative convergence.

The grid near the duct, which uses both structured and unstructured elements, is depicted in Fig.3. The average nondimensional wall distance of the first grid point in the boundary layer mesh was $y_1^+ \approx 1$ and the aspect ratio of the first element on the airfoil was set to about 20. The thickness of the boundary layer mesh was set equal to $\min(\delta, 0.95\Delta n/c)$ where $\delta$ is the thickness of the boundary layer over the airfoil estimated from the flat-plate correlation $\delta = 1.1\frac{0.16c}{Re_c^{1/7}}$, where the 1.1 factor accounts for the longer curved surface of the airfoil compared to its chord (White, 2006). This prevents the actuator disc from penetrating the boundary layer mesh which can result in the failure of boundary layer mesh generation. The fan boundary condition in Fluent requires identifying the direction of positive pressure jump. The use of triangular elements as shown in Fig. 3 between the boundary layer mesh and the outside structured mesh, made the grid generation more efficient. However, with the triangular elements used on the fan boundary condition, the specified direction of the fan boundary condition randomly changed from case to case and sometimes the actuator disc became undesirably distorted. To fix this issue, a thin structured quadrilateral grid was created just downstream of the fan boundary which is seen in Fig. 3. The width and cell-sizing of this thin grid are scaled with $c/D$ to prevent large cells near the boundary layer mesh for smaller ducts.

Two metrics were used to characterize the performance of a DWT. First, the power coefficient based on the swept area of the rotor:

$$C_P = \frac{P}{\frac{1}{2}\rho V_\infty^3 A_{rotor}} \tag{2}$$

and second, the power coefficient based on the exit area of the duct:

$$C_{P,total} = \frac{P}{\frac{1}{2}\rho V_\infty^3 A_{total}} \tag{3}$$

$C_{P,total}$ is a measure of the performance for a given total cross-sectional area of the device whereas $C_P$ is the performance for a given rotor cross-sectional area.

In order to validate the actuator disc model, the axisymmetric actuator disc without a duct was simulated in the domain shown in Fig. 2. Fig. 4 compares the axisymmetric RANS actuator disc simulation results on three different grids with the 1-D actuator disc momentum theory. The coarse, medium and fine grids had about $1.6 \times 10^4, 6.5 \times 10^4$ and $2.6 \times 10^5$ cells. The only noticeable difference between the three grids is observed at values of $C_T$ close to 1. At lower values of $C_T$ the RANS actuator disc model and the momentum theory results are visually indistinguishable. For heavily loaded actuator discs with $C_T > 1$, the simple 1-D momentum theory is not valid and empirical correlations are often used (Glauert, 1935; Sørensen et al., 1998).

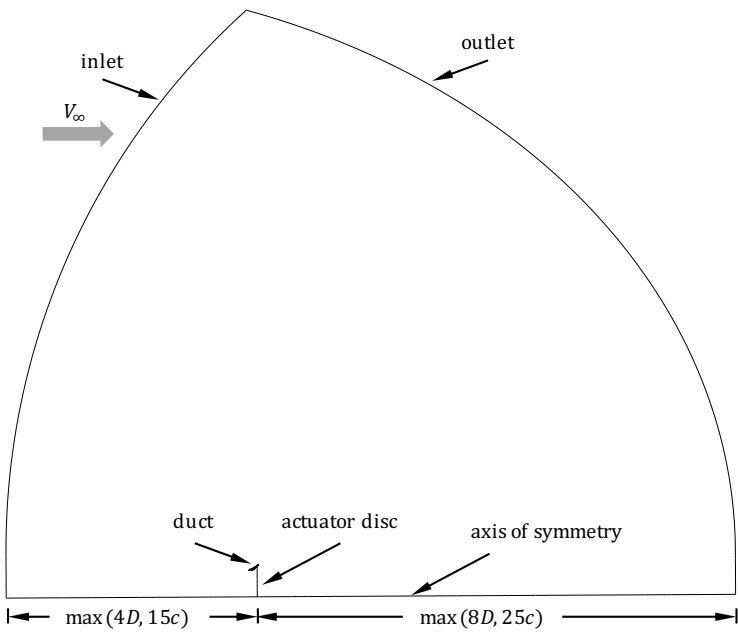

**Figure 2.** The computational domain

### 2.1 Optimization

The performance of the DWT was considered to be a function of a number of design variables, mentioned in the introduction, including the chord length of the duct cross-section, $c$, the thrust coefficient based on the axial velocity at the rotor, $C_{T,rotor}$,

145 the angle of attack of the duct cross-section, $\alpha$, the normal rotor gap, $\Delta n$, and the axial position of the rotor, $z_{rotor}$. These are shown in Fig. 1. The optimization problem was to maximize $C_{P,total}$ as a function of normalized design variables $\frac{c}{D}$, $C_{T,rotor}$, $\alpha$, $\frac{\Delta n}{c}$ and $\frac{z_{rotor}}{l}$ where $l$ is the duct length as shown in Fig. 1. Although the design variable was $C_{T,rotor}$, all the results are presented in terms of the easier to interpret thrust coefficient based on the freestream velocity $C_T = \frac{T}{\frac{1}{2}\rho V_\infty^2 A_{rotor}}$ where $T = \int_0^{D/2} 2\pi \Delta p r dr$. The constraints of the optimization were positivity of $c/D$, $C_T$, $\alpha$, and $\Delta n/c$ and $0 < z_{rotor}/l < 1$. The

150 nondimensionalization of the rotor gap by chord length instead of rotor diameter was done to help the optimization process as the optimal normal rotor gap seemed to scale with the chord length of the duct cross-section in general. The results given in the next section support this scaling.

When $c/D$ is included as a design variable, one can fix the value of $Re_c$ or $Re_D$. If $Re_D$ is fixed, the smaller values of $c/D$ result in values of $Re_c$ lower than the operating range of the airfoil for which Reynolds number dependency can be expected.

155 Additionally, the larger values of $c/D$ result in high values of $Re_c$ which require more computationally expensive boundary layer meshes. For this reason, $Re_c$ was fixed and $Re_D$ was allowed to vary. $Re_c$ was set to a value of $3 \times 10^5$. For the Eppler E423 airfoil used here, the operating range extends to $Re_c = \frac{V_\infty c}{\nu}$ as low as $1.4 \times 10^5$. For lower Reynolds numbers, large flow

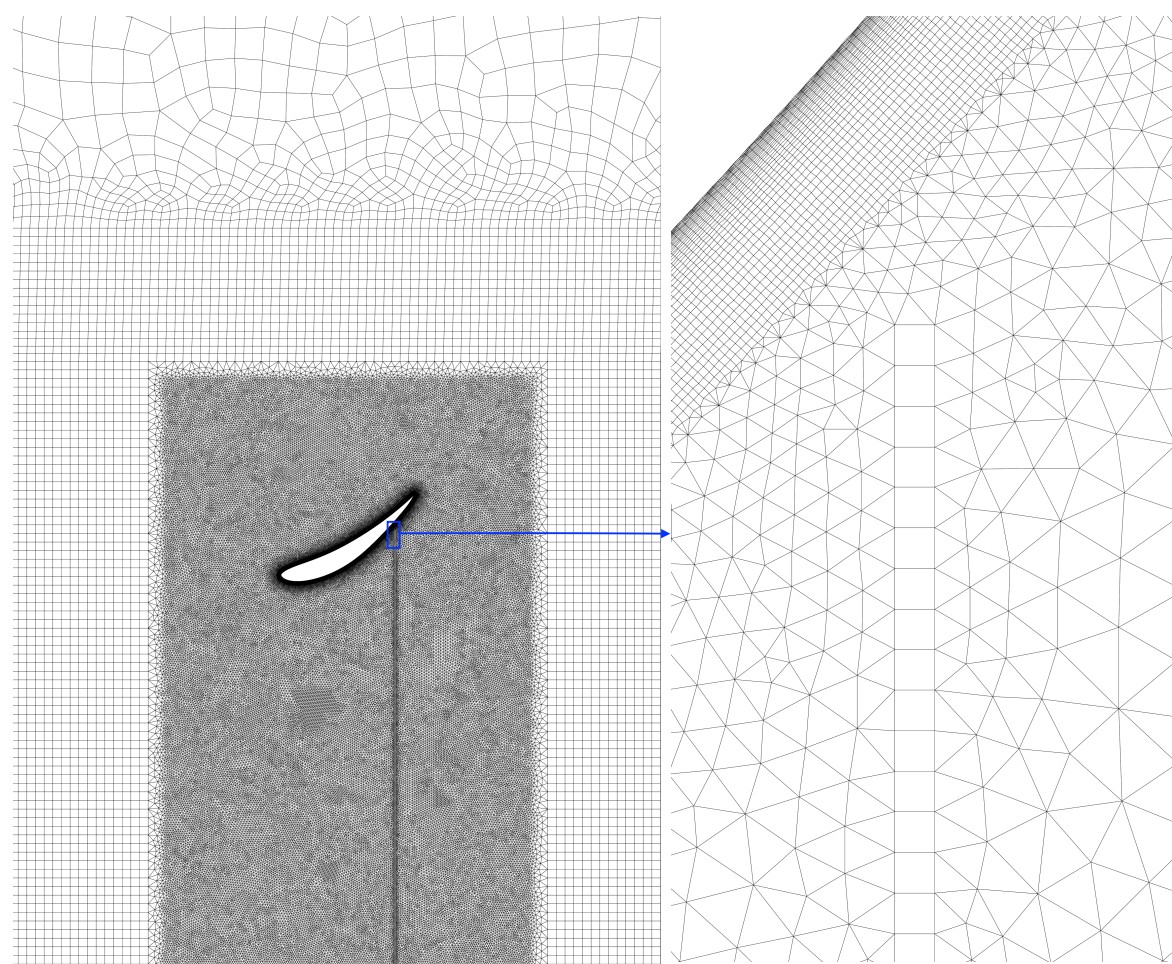

**Figure 3.** Grid near the duct and actuator disc

separation is observed before the airfoil can attain its design maximum lift coefficient (Selig et al., 1996; Jones et al., 2008). For the range of chord lengths studied, $Re_D$ varied from $6.0 \times 10^5$ to $6.0 \times 10^6$. The axisymmetric actuator disc model used here is insensitive to the Reynolds number once $Re_D$ is greater than 2000 (Sørensen et al., 1998; Mikkelsen, 2004). Within this range of $Re_D$, the value of $C_P$ only changed by about 0.07% when using the fine grid of the actuator disc validation study at $C_T = 8/9$. Therefore, fixing $Re_c$ should minimize any Reynolds number effects and keep the computational expense of cases with larger $c/D$ manageable. To determine Reynolds number sensitivity, the optimization was repeated at $Re_c = 1.2 \times 10^6$ using the optimal design at $Re_c = 3 \times 10^5$ as the starting point.

The Hooke and Jeeves direct search optimization technique (Hooke and Jeeves, 1961; Kelley, 1999; Kolda et al., 2003) was employed in this study. Our optimization study (Bagheri-Sadeghi et al., 2018) concluded that the flow inside the duct of an optimal design for either $C_P$ or $C_{P,total}$ as the objective function is on the verge of separation. The flow separation

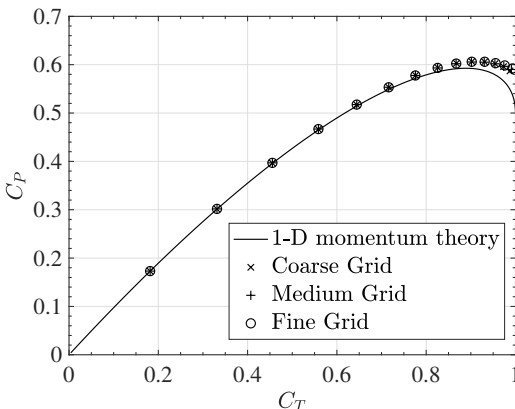

**Figure 4.** Comparison of axisymmetric RANS actuator disc model with 1-D momentum actuator disc theory

inside the duct can result in a significant drop in power output and therefore the objective function can be considered almost discontinuous close to such optimal design points. The performance of optimization methods that rely on objective function gradients is affected by the presence of such a discontinuity. The Hooke and Jeeves method, however, is less affected by such discontinuities as it does not assume a smooth objective function and only uses the objective function values to identify if a better design point is found or not.

The optimization technique starts by modifying the initial design, one design variable at a time. These exploratory moves in the design space are called steps in coordinate directions. All the design variables were scaled by their initial values and the initial step size in each coordinate direction was set to 5%. Based on the success or failure of these steps in the coordinate directions of the five-dimensional design space, the algorithm creates pattern directions, moves the base point, and increases or decreases the step sizes. The stopping criterion was $\sum_{i=1}^{N} \frac{1}{N} |\frac{\Delta x_i}{\Delta x_{i,0}}| < 0.1$ where $\Delta x_i$ is the step size in each coordinate direction, $\Delta x_{i,0}$ is the initial step size, and $N$ is the number of design variables. This means that the initially small step sizes should on average reduce by an order of magnitude for the optimization to stop. Fig. 5 shows the convergence of the optimization technique towards the optimal design. The optimization was repeated with a different initial design and the optimization approached the same optimal design point.

## 3  Results and Discussion

The optimization was performed at a constant $Re_c = 3 \times 10^5$ as described in the Methods section. The optimal design achieved $C_{P,total} = 0.69$ at $c/D = 0.18$ which corresponds to $l/D = 0.15$. The values of design variables and duct length at this optimal design are given in the second row of Table 1. The optimal design was also simulated on a finer grid with about $1.11 \times 10^6$ cells compared to $2.8 \times 10^5$ cells of the original grid. There was only a 0.03% difference between $C_{P,total}$ values. Also, simulating

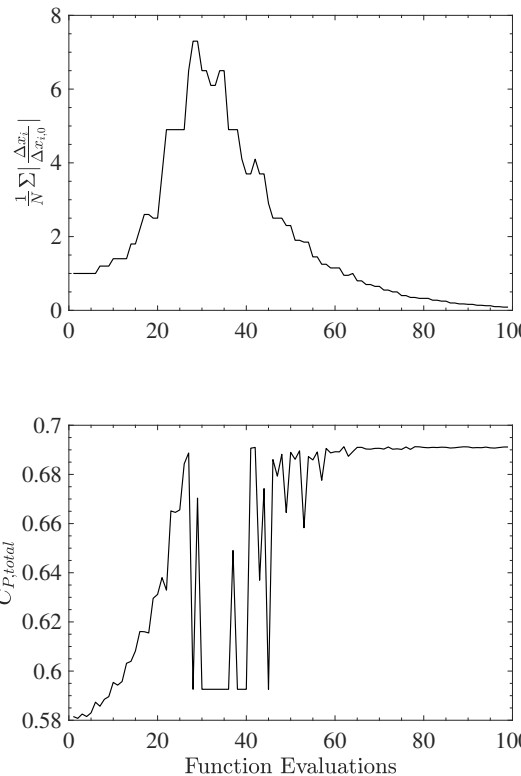

**Figure 5.** Convergence of the optimization technique to the optimal design

**Table 1.** The design variables for optimal $C_{P,total}$.

| $c/D$ $(l/D)$ | $C_T$ | $\alpha$ | $\Delta n/c$ $(\Delta n/\delta_{rotor})$ | $z_{rotor}/l$ | $C_{P,total}$ | $C_P$ |
|---|---|---|---|---|---|---|
| $0.05^\dagger$ (0.035) | 0.93 | 46° | 0.044 (1.61) | 0.97 | 0.67 | 0.69 |
| $0.18^\ddagger$ (0.15) | 0.91 | 31° | 0.040 (1.62) | 0.85 | 0.69 | 0.81 |
| $0.35^\dagger$ (0.31) | 0.92 | 25° | 0.026 (1.26) | 0.68 | 0.67 | 1.00 |
| $0.5^\dagger$ (0.47) | 0.92 | 20° | 0.024 (1.23) | 0.65 | 0.64 | 1.06 |

† The optimization performed at fixed $c/D$ at $Re_c = 3 \times 10^5$.

‡ The optimization performed with $c/D$ as a design variable at $Re_c = 3 \times 10^5$.

on a larger domain extending 1.4 times further in each direction, the value of $C_{P,total}$ changed by about -1% which was considered sufficiently accurate for the optimization study carried out here.

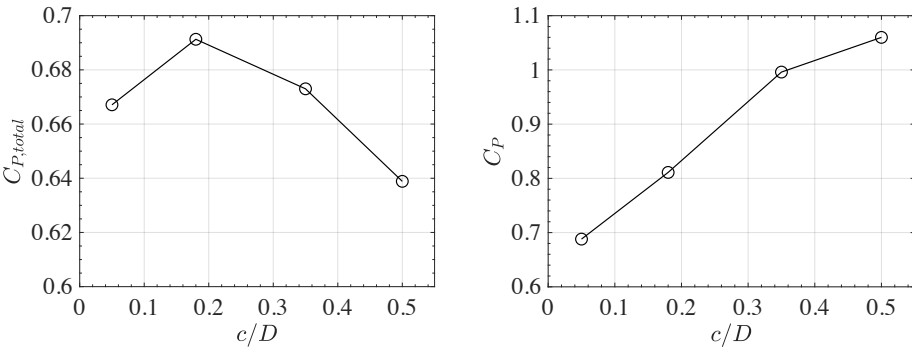

**Figure 6.** The power coefficients of designs for optimal $C_{P,total}$

To verify that there is indeed a maximum in $C_{P,total}$ with duct length, optimizations were carried out at several other fixed
$c/D$ values at $Re_c = 3 \times 10^5$. The optimal designs at these fixed values of $c/D$ are shown in Table 1 as well. Fig. 6 shows the
power coefficients of the designs for optimal $C_{P,total}$. These additional optimization studies confirm that there is an optimal
duct length for a DWT that maximizes $C_{P,total}$ and that this maximum is greater than the Betz-Joukowsky limit. Thus, for
applications desiring the greatest power per unit device area, the duct length should be around 15% of the rotor diameter. As
the results here are specific to Eppler E423, further studies are needed to verify this conclusion. The values of power coefficient
based on the rotor swept area, $C_P$ are also shown in Fig. 6 for the designs of Table 1. The values of $C_P$ seem to increase almost
linearly with $c/D$. This suggests that significantly larger values of $C_P$ can be obtained by increasing the duct length, but this
will result in a lower $C_{P,total}$.

Both here and in Bagheri-Sadeghi et al. (2018) the values of optimal $C_T$ were close to 0.9. For all optimal designs of Table
1, the value of the thrust coefficient is close to $C_T = 8/9$ which is the optimal thrust coefficient of open wind turbines at the
Betz-Joukowsky limit. This is in agreement with the momentum analysis and CFD studies of DWTs which conclude that the
optimal $C_T$ of open and ducted wind turbines are similar (van Bussel, 2007; Jamieson, 2008; Bagheri-Sadeghi et al., 2018).

The geometry and streamlines of the first three configurations in Table 1 are shown in Fig. 7b- Fig. 7d along with those for
an open rotor (in Fig. 7a). The optimal angle of attack of the duct cross-section decreased with increasing duct length. For the
open rotor, the streamlines close to the tip are at an angle of about $30°$. Furthermore, at $c/D = 0.05$ almost all of the duct cross-
section can be considered to be in the vicinity of the strong divergence of streamlines close to the rotor tip. This explains why
for the small $c/D = 0.05$ the flow is still attached at $\alpha = 46°$. The presence of the small duct with $z_{rotor}/l = 0.98$ adds extra
suction inside the duct without significantly increasing the total area of the device and achieves about 10% more power per unit
device area than an open rotor (i.e. 10% more than the optimal power output of an actuator disc RANS simulation, which was
about 0.6 as shown in Fig. 4). For the optimal design with a variable $c/D$ (i.e. $c/D = 0.18$), the increase in $C_{P,total}$ is 15%
over an open rotor and the angle of attack is reduced to $31°$. Note that the angle of streamlines of the actuator disc with respect
to horizontal decreases further downstream as the rotor wake recovers in Fig. 7. Similarly, further upstream of the rotor for the

actuator disc case, the angle of streamlines decreases. At greater duct lengths, the airfoil of the duct cross-section becomes less influenced by the presence of the actuator disc and hence the maximum $\alpha$ without a large flow separation decreases. This trend continues for $c/D = 0.5$ (see Table 1).

The optimal normal rotor gap, $\Delta n/c$ decreased with increasing duct length, but the ratio of the rotor gap to the estimated boundary layer thickness at the rotor, $\Delta n/\delta_{rotor}$, stayed nearly constant. A smaller rotor gap means a smaller exit area of the duct and therefore it helps to increase $C_{P,total}$. On the other hand, the presence of the rotor gap results in an annular jet of high-velocity air which imparts momentum to the boundary layer and helps the flow stay attached. A too small or too large $\Delta n$ weakens this annular jet. The jet is easier to see in the contour plot of $c/D = 0.35$ in Fig. 7d. For all of the optimal cases shown

in Table 1, $\Delta n/\delta_{rotor}$ was close to 1. When the chord is small relative to $D$, the dominant length scale is $c$. This determines the flow scales as well as the boundary layer thickness. Also, note that because $Re_c$ is held constant, the boundary layer thickness at the trailing edge $\delta$ scales linearly with $c$. Therefore, as optimal $\Delta n/\delta_{rotor}$ is nearly constant, optimal $\Delta n/c$ only slightly decreases as the optimal rotor position is moved further upstream resulting in a smaller boundary layer thickness at the rotor, $\delta_{rotor}$ compared to the trailing edge. This justifies the scaling of $\Delta n$ by the chord length of the duct.

Similar to Bagheri-Sadeghi et al. (2018), the design for optimal $C_{P,total}$ resulted in downstream placement of the rotor. The optimal rotor position moved further upstream in the duct as $c/D$ increased. Note that at greater duct lengths the annular jet is stronger as can be seen in Fig. 7. This further upstream placement of the rotor means that the annular jet formed between the rotor tip and duct can exchange momentum with a larger portion of the boundary layer, which may help the DWT attain more power per total unit device area without flow separation.

The sensitivity of the total power coefficient of the optimal design to different design variables $x_i$, is shown in Fig. 8 . The greatest sensitivity is to the thrust coefficient of the rotor which matches the results of previous studies (Venters et al., 2018; Bagheri-Sadeghi et al., 2018) and illustrates the importance of rotor design in achieving optimal power output from a DWT (A rotor design approach based on the blade element momentum method using axisymmetric RANS actuator disc simulations as input is discussed by Kanya and Visser (2018)). For $\alpha$ and $z_{rotor}/l$, an increase in the design variable causes flow separation

inside the duct and a significant drop in the power output. This is demonstrated by an increased sensitivity of $C_{P,total}$ to increase in $\alpha$ and $z_{rotor}/l$ compared to their reduction. The sensitivity to the reduction in $z_{rotor}/l$ is mainly driven by changes in the total area of the device rather than changes in the power output. Therefore as concluded in Bagheri-Sadeghi et al. (2018) a smaller DWT with similar power output (i.e. a greater $C_{P,total}$) can be designed by placing the rotor further downstream of the duct. The observation that the optimal design is on the verge of flow separation with respect to $\alpha$ agrees with the results

of Bagheri-Sadeghi et al. (2018) obtained for the design for optimal $C_P$. However, for the design for optimal $C_P$ in Bagheri-Sadeghi et al. (2018) the flow is on the verge of separation with respect to rotor gap and shows little sensitivity to $z_{rotor}/l$, whereas the optimal design for $C_{P,total}$ shows the least sensitivity in Fig. 8 to $\Delta n/c$ whereas a slight increase in $z_{rotor}/l$ leads to flow separation.

    Fig. 9 compares the $C_{P,total}$ vs $C_T$ curves of the optimal DWT with the RANS actuator disc model. Note that for an open

rotor turbine $C_{P,total} = C_P$. Close to the optimal design, at a given $C_T$ the DWT produces about 15% more power per unit device area. However, at lower values of $C_T$, this increase in $C_{P,total}$ becomes smaller (e.g. at $C_T \approx 0.55$ the DWT produces

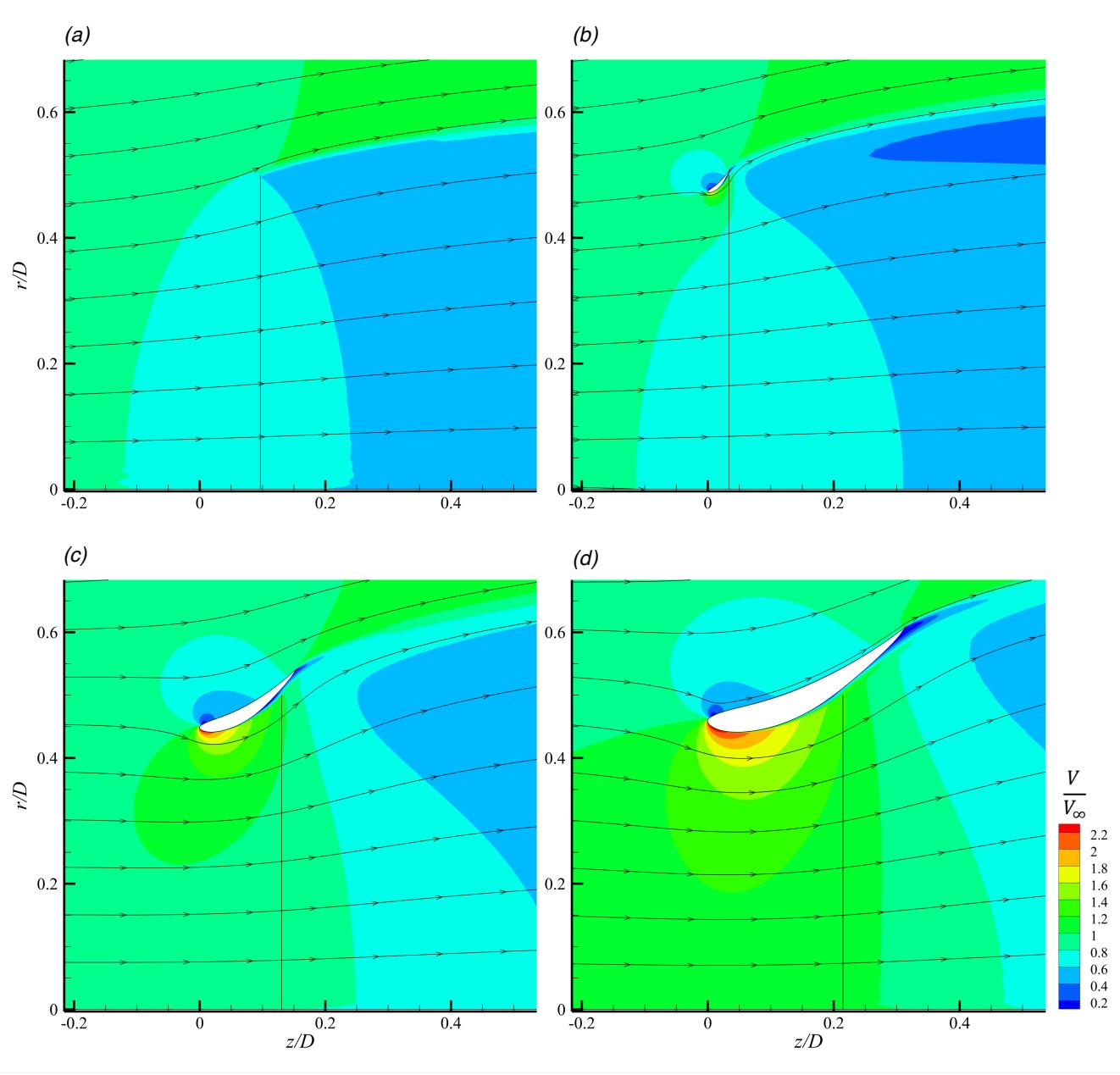

**Figure 7.** The streamlines and contours of nondimensional velocity magnitude $V/V_\infty$ of (a) actuator disc, shown by the radial black line, at $C_T = 8/9$, and the designs for optimal $C_{P,total}$ at $Re_c = 3 \times 10^5$ at (b) $c/D = 0.05$, (c) $c/D = 0.18$, and (d) $c/D = 0.35$

only 6% more power per total device area). In other words, at lower rotor loadings, the power output of the optimal DWT per total device area approaches that of an open rotor turbine. Generally, when using actuator disc simulations, the increase

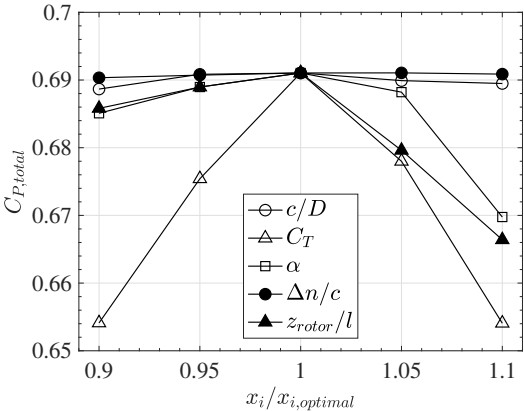

**Figure 8.** The sensitivity of the total power coefficient of the optimal design to different design variables

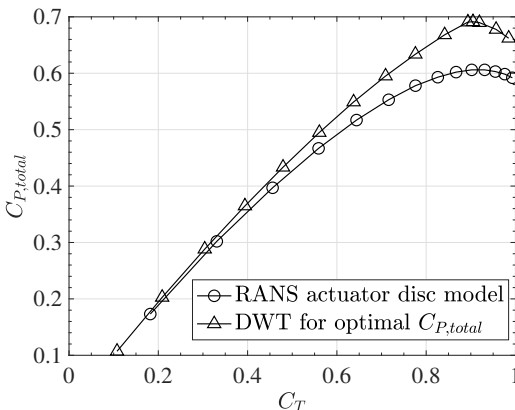

**Figure 9.** Comparison of $C_{P,total}$ vs $C_T$ for RANS actuator disc model and the DWT for optimal $C_{P,total}$

in velocity of a DWT compared to an open wind turbine should be fixed so the ratio of $C_P$ values (and therefore $C_{P,total}$ values in Fig. 9 as the $A_{total}$ is fixed) should be independent of $C_T$ (Hansen et al., 2000). Indeed, if the curves of Fig. 9 were plotted for another DWT design not on the verge of flow separation (e.g. the optimal design but at a reduced $\alpha$) the ratio of the power coefficients would stay constant. The constant ratio of the power coefficients is not seen here at the optimal design because the optimal design is on the brink of flow separation and at lower values of $C_T$ the rotor is less effective in keeping the flow attached and the induced velocity decreases. As the thrust coefficient reduces, the high-speed annular jet becomes weaker and therefore cannot keep the flow attached and gradually the region of flow separation extends further upstream. The flow separation with thrust coefficient reduction is gradual rather than the abrupt separation which can be observed by slightly increasing the angle of attack or $z_{rotor}/l$ in Fig. 8.

**Table 2.** The Reynolds number sensitivity of optimal design for $C_{P,total}$.

| $c/D$ ($l/D$) | $C_T$ | $\alpha$ | $\Delta n/c$ ($\Delta n/\delta_{rotor}$) | $z_{rotor}/l$ | $C_{P,total}$ | $C_P$ |
|---|---|---|---|---|---|---|
| $0.18^{\dagger}$ (0.15) | 0.91 | 31° | 0.040 (1.62) | 0.85 | 0.69 | 0.81 |
| $0.18^{\ddagger}$ (0.15) | 0.90 | 33° | 0.040 (1.97) | 0.85 | 0.70 | 0.83 |

† The optimization performed with $c/D$ as a design variable at $Re_c = 3 \times 10^5$.
‡ The optimization performed with $c/D$ as a design variable at $Re_c = 1.2 \times 10^6$.

## 3.1 Reynolds Number Sensitivity

To examine Reynolds number sensitivity, the simulation of the optimal design was repeated at $Re_c = 1.2 \times 10^6$. The Reynolds
numbers used in this study correspond to typical values for DWTs, which typically target residential applications. For example,
at a wind speed of $V_\infty = 10$ m/s at standard sea-level temperature and pressure, the $Re_c = 3 \times 10^5$ design corresponds to
$c = 0.44$ m, $D = 2.4$ m and $P = 2.3$ kW, and $Re_c = 1.2 \times 10^6$ corresponds to $c = 1.76$ m, $D = 9.8$ m and $P = 38$ kW.

The result of optimization at this higher Reynolds number is shown in the second row of Table 2. The maximum $C_{P,total}$
slightly increased to 0.70. The optimal value of $c/D$ did not change to the precision of the optimization. The optimal values of
$\Delta n/c$ and $z_{rotor}/l$ also did not change and the optimal $C_T$ and $\alpha$ only varied slightly. This confirms that at $Re_c = 3 \times 10^5$ the
Reynolds number dependency was small.

The contour plot of eddy viscosity ratio $\frac{\mu_t}{\mu}$ for the optimal design is shown in Fig. 10a. On the suction side of the duct at
$Re_c = 3 \times 10^5$ the turbulence model is activated near the leading edge of the duct. This suggests that the optimal design should
be insensitive to Reynolds number. The eddy viscosity ratio at $Re_c = 1.2 \times 10^6$ is shown in Fig. 10b. At this Reynolds number
the turbulence model is activated slightly further upstream and the power output increased to $C_P = 0.82$. The small increase
can be explained by the reduced flow separation near the trailing edge at this greater Reynolds number, which increased the
effective exit area of the duct. Also, note that the turbulence model on the pressure side becomes activated further upstream as
well. However, this should not matter in identifying the optimal design as the flow stays attached on the outside of the duct.
The $k - \omega$ SST turbulence model, with the apparent transition near the leading edge, indicates that the flow is entirely turbulent
over the airfoil, and therefore the results should not change significantly even at higher Reynolds numbers.

## 4 Conclusions

The optimal design of a ducted wind turbine with the Eppler E423 airfoil as its cross-section was investigated using CFD
simulations of axisymmetric RANS equations with the $k - \omega$ SST turbulence model and an actuator disc. The total power
coefficient characterizing the power output per total area of the device, $C_{P,total}$, was used as the design objective. The design
variables included the chord length and the angle of attack of the duct cross-section, the thrust coefficient, the rotor axial
position, and the tip clearance of the rotor.

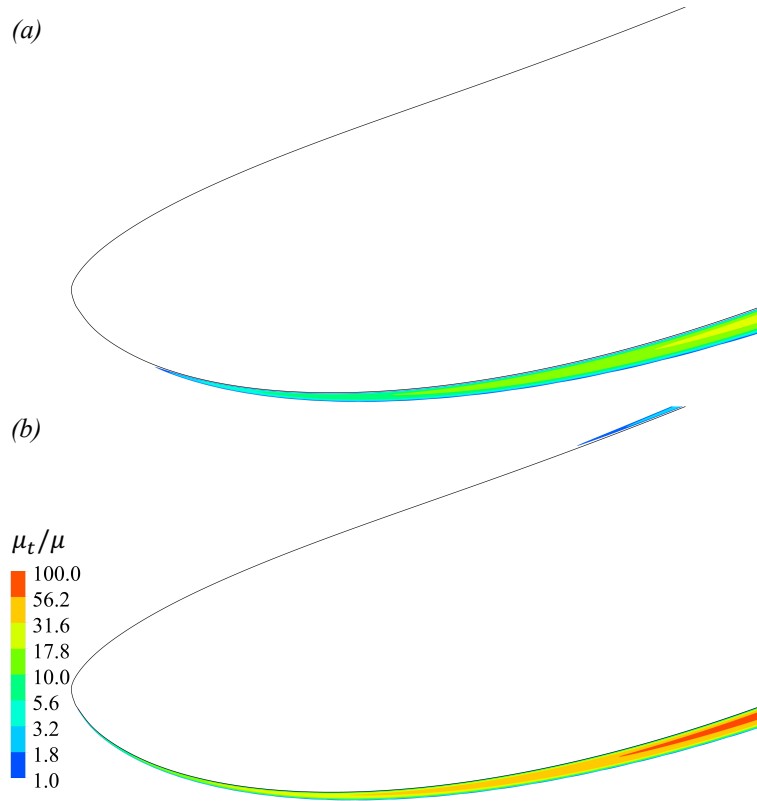

**Figure 10.** Eddy viscosity ratio contours near the leading edge of the duct for the optimal design ($c/D = 0.18$, $C_T = 0.91$, $\alpha = 31°$, $\Delta n = 0.040$, $z_{rotor}/l = 0.85$) at (a) $Re_c = 3 \times 10^5$ (b) $Re_c = 1.2 \times 10^6$

The results demonstrate the existence of a maximum $C_{P,total}$ for ducted wind turbines which sets an upper limit for ducted wind turbines similar to the Betz-Joukowsky limit for open rotors. For the Eppler E423 airfoil, this maximum was obtained for a duct length of about 15% of the rotor diameter. With this duct length a $C_{P,total}$ of 0.70 was obtained which exceeds what can be obtained with an open rotor by 16%. This is the first time that an optimal duct length has been identified, although the optimization was for $C_{P,total}$ not $C_P$. The value of $C_P$ increased almost linearly with duct length over the range investigated.

Additionally, the results of optimization at fixed $c/D$, suggested that the optimal value of the design variables can significantly change with the duct length. In agreement with previous theoretical and numerical studies, for all duct lengths the optimal thrust coefficient, $C_T$, was almost 0.9 which is similar to open rotor turbines. The results also showed that the optimal design for $C_{P,total}$ was most sensitive to the thrust coefficient of the rotor, which indicates the importance of proper rotor design. At lower than optimal thrust coefficients, the power per unit device area, $C_{P,total}$, of the optimal DWT design gradually approached that of an open wind turbine. This can be explained by considering that the optimal design was on the verge of flow separation and the rotor became less effective in keeping the flow attached as the thrust coefficient decreased.

The optimal angle of attack of the duct cross-section decreased significantly with increasing the duct length. Additionally, the optimal design was on the verge of flow separation with respect to the angle of attack of the duct cross-section and the axial position of the rotor.

The optimal normal rotor gap was close to the boundary layer thickness at the rotor. Therefore, the optimal normal rotor gap scaled proportional to the chord length of the duct cross-section as the turbulent boundary layer thickness almost linearly increases with the chord length of the duct cross-section. This gap is needed to create the high-velocity annular jet, which helps keep the boundary layer attached.

The optimal rotor position was at the rear of the duct but at greater values of duct length moved further upstream in the duct. This further upstream position was more effective at eliminating flow separation and hence allowed greater values of $C_{P,total}$.

*Data availability.* Data available upon request from the corresponding author.

*Author contributions.* NB contributed to the methodology, ran the simulations and post-processed the data and wrote the first draft of the paper. BH supervised the study and contributed to the conceptualization, methodology, writing and revision of the paper. KV contributed to the conceptualization and revision of the paper.

*Competing interests.* The authors declare no conflict of interests.

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
