# Peer review of "Maximal power per device area of a ducted turbine"

_Wind Energy Science, 2021_

## Referee Comment (RC1)

This paper is a useful addition to the literature on diffuser-augmented wind turbines (DAWTs) addressing the issues of maximum performance and appropriate duct length, which are of major practical importance. The computational mesh shown in figure 3 looks to be of high quality and the grid independence study confirms the validity of the results.

The recent review by Bontempo & Manna (2020, additional references below) concluded that the maximum performance of a DAWT is an open question still and did not seriously address the issue of duct length. It has been argued, at least as far back as Lubitz & Shomer (2014), that DAWT performance based on maximum duct area was no better than that of a bare turbine which is subject to the Betz-Joukowsky limit, Okulov & van Kuik (2012). Further numerical and experimental data is analyzed by Limacher et al. (2020). The present paper suggests that a modest improvement is possible with an airfoil-shaped, but surprisingly short, duct.

It has also become fashionable to use a duct shape that is simpler in cross-section than an airfoil, and an exit flange. Limacher et al. (2020) show that this arrangement is not optimal and that a "lifting" duct is needed. Hjorst & Larsen (2014) considered a multi-element duct which would allow re-energizing of the boundary layer and I would like to see a discussion of relative merits of this and a single element. Other minor comments are:

1. pattern search methods have developed since Hooke & Jeeves (1961, reference in paper). For example, the methods used in Matlab were published at the end of the 20th century. The optimization method does not influence the accuracy of the individual simulations but the issue of whether an optimum is obtained is a difficult one.

2. I take it that the thin radial line in figure 7 is the actuator disk. This should be stated in the figure caption.

3. Given the importance of the duct length and the lack of information on its optimal value in the literature, I was surprised that it was not mentioned in the conclusions. I suggest that the authors remedy this deficiency.

Additional References

Bontempo, R., & Manna, M. (2020). Diffuser augmented wind turbines: Review and assessment of theoretical models. Applied Energy, 280, 115867.

Hjort S, Larsen H. A multi-element diffuser augmented wind turbine (2014). Energies,7(5):3256–81.

Limacher, E. J., da Silva, P. O., Barbosa, P. E., & Vaz, J. R. (2020). Large exit flanges in diffuser-augmented turbines lead to sub-optimal performance. Journal of Wind Engineering and Industrial Aerodynamics, 204, 104228.

Lubitz, W.D., Shomer, A. (2014). Wind loads and efficiency of a diffuser augmented wind turbine (DAWT). Proc. Can. Soc. Mech. Eng. Int. Cong. 1–5, 2014.

Okulov, V.L., van Kuik, G.A.M. (2012). The Betz–Joukowsky limit: on the contribution to rotor aerodynamics by the British, German and Russian scientific schools. Wind Energy 15 (2), 335–344.

---

## Referee Comment (RC3)

**Supplementary comments**

There is a lot of food for thought in;

https://www.researchgate.net/publication/344364853 Rethinking Ducted Turbines The Fundame ntals of Aerodynamic Performance and Theory

Figure 7.4: Optimum rotor loading for 1776 aerofoil and diffuser parameter duct shapes.

The value of 8/9 is only approached as the area ratio tends to 1 and length to zero equivalent to having no duct – open flow - and ducts that may be considered useful did not maximise Cp at Ct above ~ 0.84.

For a while I thought that a fully optimised finite duct in inviscid flow would achieve Cp max at Ct = 8/9. I was able to construct an analytic potential flow model of an ideal planar duct of infinite length which always maximised Cp at Ct =8/9 regardless of area ratio. However I now suspect, as suggested by the figure above, that no finite duct, in inviscid flow, will maximise rotor Cp with a loading as great as Ct=8/9.

What is very clear and has been for a long time is that the optimal loading of a rotor in a duct is not far different from in open flow. I believe that experimental results (Foreman etc) in support of 8/9 and experimental values above 8/9 around unity (Phillips) arise from viscous flow effects that energy in some of the flow that does not pass through the duct can assist flow through the duct. I refer here to "simple" ducts without slotted aerofoils of boundary layer injection which obviously exploits external flows.

The most efficient ducts, as measured by rotor Ct which is identically equal to the fraction of source kinetic power extractable at the rotor plane, are aerofoil ducts but, in inviscid flow, ducts for a given rotor size and specific area ratio that maximise rotor Cp are right angle ducts. Ct at Cp max may be below 0.8 with these ducts, implying that they extract source energy less efficiently, but they maximise Cp overall by inducing flow from a relatively larger source area than the more efficient aerofoil ducts. There is some experimental evidence of this from data on the WindLens duct (Jamieson, "Innovation of Wind Turbine Design" 2nd edition 2018) and results from this lead me to think that the most effective real ducts will combine some aerofoil shape expansion with a more abrupt exit flange.

---

## Author Comment (AC1)

**Reply to Comments by Prof. David Wood (Reviewer 1)**

Thank you for going through the paper and providing helpful feedback. Our responses to your comments are given below, where your comments are italicized and our responses are in plain text.

*This paper is a useful addition to the literature on diffuser-augmented wind turbines (DAWTs) addressing the issues of maximum performance and appropriate duct length, which are of major practical importance. The computational mesh shown in figure 3 looks to be of high quality and the grid independence study confirms the validity of the results. The recent review by Bontempo & Manna (2020, additional references below) concluded that the maximum performance of a DAWT is an open question still and did not seriously address the issue of duct length. It has been argued, at least as far back as Lubitz & Shomer (2014), that DAWT performance based on maximum duct area was no better than that of a bare turbine which is subject to the Betz-Joukowsky limit, Okulov & van Kuik (2012). Further numerical and experimental data is analyzed by Limacher et al. (2020). The present paper suggests that a modest improvement is possible with an airfoil-shaped, but surprisingly short, duct. It has also become fashionable to use a duct shape that is simpler in cross-section than an airfoil, and an exit flange. Limacher et al. (2020) show that this arrangement is not optimal and that a "lifting" duct is needed. Hjorst & Larsen (2014) considered a multi-element duct which would allow re-energizing of the boundary layer and I would like to see a discussion of relative merits of this and a single element.*

Thanks for the suggested additional references. The reference to Betz limit was changed to Betz- Joukowsky limit. The suggested references were added to the revised manuscript. Also, a brief discussion of multielement and slotted ducted wind turbines was added to the introduction as well:

Bontempo and Manna (2020) reviewed various theoretical models of ducted

wind turbines and concluded that they are all equivalent and that their apparent differences are due to different choices of flow parameters used to characterize the effect of the duct (e.g. the exit pressure coefficient (Foreman et al. 1978) or extra back pressure velocity ratio (van Bussel, 2007)). They also show that these models are insufficient in predicting the optimal design of ducted turbines as they neglect the dependence of the flow parameters on the thrust coefficient. However, they demonstrated that $C_{P,total}$ greater than Betz-Joukowsky limit can be achieved at $C_T = 8/9$.

Further increases in $C_{P,total}$ could be possible by delaying boundary layer separation, e.g. by using multi-element or slotted ducts (Igra, 1981; Gilbert and Foreman, 1983; Phillips et al., 2002; Hjort and Larsen, 2014), as the optimal design tends to be on the verge of flow separation (Bagheri-Sadeghi, 2018). Large flanges are often used at the exit of ducted turbines to further lower the pressure at the exit plane of the duct and increase the swallowing effect. Limacher et al. (2020) show that large flanges lead to reduced values of $C_{P,total}$.

*Other minor comments are:*

*1. pattern search methods have developed since Hooke & Jeeves (1961, reference in paper). For example, the methods used in Matlab were published at the end of the 20th century. The optimization method does not influence the accuracy of the individual simulations but the issue of whether an optimum is obtained is a difficult one.*

The pattern search methods used in MATLAB are based on the generalized pattern search method [1, 2]. The Hooke and Jeeves method is a special case of this generalized pattern search method [1]. Torczon [1] establishes the global convergence of these methods for continuously differentiable functions.

Still, it is difficult to ensure the global optimal point is reached, but for two reasons our results indicate that our simulations reached an optimal design at a duct length of 15% of rotor diameter. First, we repeated the optimization (with the chord length of the duct as a design variable) from two different initial designs and the optimization scheme ended up at the same optimal design point within the precision of the optimization. Second, when we did optimization at three fixed chord lengths of the duct cross-section about the optimal design, the objective function decreased from its optimal value as shown in Fig. 6.

The Hooke and Jeeves method is a classic direct search method that is simple, effective and robust and therefore remains popular for simulation-based optimization studies [2]. For example, it is one of the optimization algorithms included in the iSight optimization package used for simulation-based optimization [3]. [4] compares the performance of three popular direct search methods including Hooke and Jeeves, multidirectional search [5] and Nelder-Mead's simplex algorithm [6] for several example problems including cases of a perturbed quadratic function and Weber's problem, which is not differentiable at its minimum, and in both cases Hooke and Jeeves method performs better or as well as other methods. However, the Hooke and Jeeves method is inherently sequential and other derivative-free methods should be preferred for simulation-based optimization problems with large number of design variables where parallelization is desirable.

*2. I take it that the thin radial line in figure 7 is the actuator disk. This should be stated in the figure caption.*

The caption now mentions that the radial black line is the actuator disc.

*3. Given the importance of the duct length and the lack of information on its optimal value in the literature, I was surprised that it was not mentioned in the conclusions. I suggest that the authors remedy this deficiency.*

The existence of an optimal duct length is now further emphasized in the conclusions:

> This is the first time that an optimal duct length has been identified, although the optimization was for $C_{P,total}$ not $C_P$. The value of $C_P$ increased almost linearly with duct length over the range investigated.

**References**

[1] V. Torczon, "On the convergence of pattern search algorithms," *SIAM Journal on Optimization*, vol. 7, no. 1, pp. 1–25, 1997.

[2] T. G. Kolda, R. M. Lewis, and V. Torczon, "Optimization by direct search: New perspectives on some classical and modern methods," *SIAM Review*, vol. 45, no. 3, pp. 385–482, 2003.

[3] A. Van der Velden and P. Koch, "Isight design optimization methodologies," *ASM handbook*, vol. 22, p. 79, 2010.

[4] C. T. Kelley, *Iterative Methods for Optimization.* Society for Industrial and Applied Mathematics, 1999.

[5] J. E. Dennis, Jr. and V. Torczon, "Direct search methods on parallel machines," *SIAM Journal on Optimization*, vol. 1, no. 4, pp. 448–474, 1991.

[6] J. A. Nelder and R. Mead, "A Simplex Method for Function Minimization," *The Computer Journal*, vol. 7, pp. 308–313, 01 1965.

**Reply to Comments by Reviewer 2**

Thank you for going through the paper and providing helpful feedback. Our responses to your comments are given below, where your comments are italicized and our responses are in plain text.

*The work is an optimization of a shrouded wind turbine using axisymmetric CFD, where the rotor is modeled as an actuator disc. The parameters are the non-dimensionalized length and diameter of the diffusor (based on an Eppler E423 airfoil ) and the inflow angle. The k-omega, SST turbulence is used and is considered a good choice. However, the Re number based on the chord is only 300.000 which is quite low for this turbulence model and where transition is very important. Why this low Re number and could a transition model such as gamma-Retheta be applied ? It could be because it is intended for small rotor, and are there any experimental data available ?*

A discussion of Reynolds number dependency was added to the revised manuscript. Also, as our goal was to obtain an upper limit for performance of ducted wind turbines with minimal Reynolds number dependency rather than the performance at a specific Reynolds number, therefore a transition model like $\gamma - Re_\theta$ which depends strongly on the turbulence intensity and Reynolds number was not used.

*A constant local thrust is specified in Eq. 1 and the rotor is modelled as a pressure jump. Would it be possible to also include a tangential load creating swirl that may also change the pressure ? Instead of using simply the pressure one could have used volume forces that would allow later to model a real blade geometry. Is a constant CT the optimum for a shrouded WT, where the inflow velocities are high near the shroud airfoil ? It seems that by putting the rotor further back as shown in Figure 7 the flow is quite constant.*

The thrust coefficient was actually changing in the radial direction which means the pressure drop was larger near the tip where velocities are greater. This is now stated more clearly in the methods section. Including real lift and drag airfoil data, as in a combined CFD-blade element design method, although interesting and useful in

simultaneously optimizing the design of rotor blades and duct, was not really the goal of this study. The actuator disc model was used to give an upper limit for what can be obtained using ducted wind turbines. The goal of our work was to assess this upper limit rather than modeling or optimizing the rotor blade geometry.

The effect of swirl is more complex. Although including the effect of wake rotation by allowing a swirl component leads to power coefficients lower than that of a simple actuator disc, the interaction of the swirl component with the boundary layer near the walls of the duct may delay the boundary layer separation and lead to a design with a slightly greater power coefficient. Although we do not believe that the inclusion of the swirl component in the actuator disc model would significantly change the results of the optimization study, the effects of inclusion of the swirl component of wake in our model is an interesting topic and is left for our future studies.

*The grid is very coarse, in the order 0.2-1 million cells, which may be justified by the relatively low Re number. The number of parameters run in the order 5 is quite coarse but the results shown looks physical.*

A grid convergence study was performed with the results discussed in the paper that identifies the results were grid-independent (please see line 185).

**Reply to Comments by Mr. Peter Jamieson (Reviewer 3)**

Thank you for going through the paper and providing helpful feedback. Our responses to your comments are given below, where your comments are italicized and our responses are in plain text.

*This paper explores and aims to quantify a limit on power output from a ducted turbine and I agree with Reviewer RC1 is a useful contribution to DAWT literature. It is totally logical that there is a limit to power from any energy extraction system related to its dimensions and the work adds to evidence showing that such a limit does exceed the Betz limit based on the exit area (if presumed to be the maximum sectional area of the duct) and discrediting a common fallacy that may cause ducted turbine concepts to be undervalued. I think the paper is a nice piece of work taken in the specific context of the Eppler E423 aerofoil but the comment (line 80) " a similar result should hold for other duct cross sections as well" is too much of a stretch without direct evidence.*

We added some more to the introduction to emphasize the limits of the study by noting that our study does not include designs like multi-element ducts or effect of an exit flange. Also, we have mentioned several times that the results are specific to Eppler E423 airfoil (please see lines 88, 92, 192 and 284).

*I believe this paper would have been strengthened, perhaps hugely, by parallel investigations in inviscid flow. This is because in real flow, there are two quite different contributions to a power limit, one fundamentally related to device size (possibly only maximum section area) and another that may or may not be fundamental relating to flow separation effects.*

The effect of viscous effects is mainly manifested by flow separation which is accompanied by a large drop in power output of the DWT. From the sensitivity study shown in Figs. 8 and 9 the optimal design for maximum $C_{P,total}$ is on the verge of flow

separation with respect to thrust coefficient, angle of attack of the duct cross-section and the axial position of the rotor.

Therefore, including the viscous effects acts as a penalty function during the optimization process that prevents the optimization ending at an optimal design point that is not physically feasible (e.g. with a large angle of attack of the duct cross-section). Still, it is interesting to isolate the viscous effects by comparing the optimal designs from inviscid simulations and that of viscous simulations in our future work. We think that there should be at least some bounds on design variables during the optimization using inviscid simulations, e.g. with respect to the angle of attack of the cross-section, to avoid unphysical optimal designs.

*Although the references are quite full and relevant, in addition those mentioned by RC1, I think the PhD thesis of McLaren-Gow (reference attached) and some other of his publications can shed a lot of light on the present topics. I was stimulated by your paper to process some of his results on right angled ducts (cylinder with exit flange) and aerofoil ducts (represented only by the camber line profile) and perhaps see a limit on Cp total even when rotor Cp in inviscid flow is unbounded. Regarding your comments around line 40, I attach a reference with a figure and some explanation - basically in inviscid flow it seems that no finite duct will realise rotor Cp max at a Ct value as great as 8/9 but in real flow, due to some benefit from external flows, Cts associated with rotor Cp max around and a little above 8/9 can result.*

It is interesting that the viscous effects seem to be essential to realize an optimal $C_T$ of 8/9. This is now added to the introduction with reference to McLaren-Gow's work:

> However, McLaren-Gow (2020) performed axisymmetric inviscid simulations of DWTs with various duct shapes with an actuator disc and concluded that the value of $C_T$ to maximize $C_P$ is lower than 8/9.

*I find the comments on duct length very interesting and a valuable investigation as far as it goes but again find it a step too much to generalise from the one aerofoil that there is an optimum duct length 15% D. The changes I would strongly recommend are to qualify the comments in line 80 and 176.*

These generalizations are now qualified as you suggested.

*The value of 8/9 is only approached as the area ratio tends to 1 and length to zero equivalent to having no duct – open flow - and ducts that may be considered useful*

*did not maximise Cp at Ct above   0.84. For a while I thought that a fully optimised finite duct in inviscid flow would achieve Cp max at Ct = 8/9. I was able to construct an analytic potential flow model of an ideal planar duct of infinite length which always maximised Cp at Ct =8/9 regardless of area ratio. However I now suspect, as suggested by the figure above, that no finite duct, in inviscid flow, will maximise rotor Cp with a loading as great as Ct=8/9.*

*What is very clear and has been for a long time is that the optimal loading of a rotor in a duct is not far different from in open flow. I believe that experimental results (Foreman etc) in support of 8/9 and experimental values above 8/9 around unity (Phillips) arise from viscous flow effects that energy in some of the flow that does not pass through the duct can assist flow through the duct. I refer here to "simple" ducts without slotted aerofoils of boundary layer injection which obviously exploits external flows.*

*The most efficient ducts, as measured by rotor Ct which is identically equal to the fraction of source kinetic power extractable at the rotor plane, are aerofoil ducts but, in inviscid flow, ducts for a given rotor size and specific area ratio that maximise rotor Cp are right angle ducts. Ct at Cp max may be below 0.8 with these ducts, implying that they extract source energy less efficiently, but they maximise Cp overall by inducing flow from a relatively larger source area than the more efficient aerofoil ducts. There is some experimental evidence of this from data on the WindLens duct (Jamieson, "Innovation of Wind Turbine Design" 2nd edition 2018) and results from this lead me to think that the most effective real ducts will combine some aerofoil shape expansion with a more abrupt exit flange.*

The most effective DWT in most situations is probably ultimately evaluated in terms of some economic metric like the levelized cost of energy (LCOE) and a simple flanged diffuser may have a lower LCOE than a more expensive airfoil-shaped duct. However, as the study of Limacher et al. (2020) shows large exit flanges lead to suboptimal values of $C_{P,total}$ and a duct shape with only a small exit flange similar to a Gurney flap may lead to optimal $C_{P,total}$.

[revised manuscript text omitted]